# Elucidating Flavonoid and Antioxidant Activity in Edible and Medicinal Herbs *Woodwardia japonica* (L.f.) Sm. Based on HPLC-ESI-TOF-MS and Artificial Neural Network Model: Response to Climatic Factors

**DOI:** 10.3390/molecules28041985

**Published:** 2023-02-20

**Authors:** Xin Wang, Jianguo Cao, Lin Tian, Baodong Liu, Yawen Fan, Quanxi Wang

**Affiliations:** 1College of Life Science and Technology, Harbin Normal University, Harbin 150025, China; 2Heilongjiang Research Center of Genuine Wild Medicinal Materials Germplasm Resources, Harbin 150025, China; 3College of Life Sciences, Shanghai Normal University, Shanghai 200234, China; 4Changchun Institute of Technology, College of Water Conservancy and Environmental Engineering, Changchun 130012, China; 5Shanghai Key Laboratory of Plant Functional Genomics and Resources, Chinese Academy of Sciences, Shanghai Chenshan Botanical Garden, Shanghai 200234, China

**Keywords:** *Woodwardia japonica*, edible and medicinal fern, total flavonoid content, antioxidant activities, climate factors, artificial neural network model

## Abstract

*Woodwardia japonica* is a kind of great potential edible and medicinal fern. In a previous study, it was found that flavonoid and antioxidant activity of *W. japonica* from different sites were different. However, the cause of the differences has still been unclear, which has restricted the utilization of *W. japonica*. In this paper, flavonoid and antioxidant activity of *W. japonica* from nine different regions were determined with the method of a colorimetric assay with UV-VIS spectrophotometry and HPLC-ESI-TOF-MS, and the effects of climate factors on flavonoids and antioxidant activities were evaluated by mathematical modeling and statistical methods. The results showed: (1) total flavonoid content (TFC) of *W. japonica* from Wuyi Mountain (Jiangxi) was the highest, which might be related to the low temperature; (2) the differences of antioxidant activities of *W. japonica* might be related to precipitation; (3) five flavonols, two flavones and one isoflavone were tentatively identified in *W. japonica*; (4) flavonol and isoflavone might be affected by sunshine duration, and flavones were probably related to temperature. In conclusion, the effects of climate factors on flavonoids and antioxidants are significant, which would provide an important basis for further exploring the mechanism of climate affecting secondary metabolites.

## 1. Introduction

As a minority part of vegetable consumption in many communities across the globe, edible ferns are gradually attracting attention [1,2]. Young fronds, rhizomes, stems and tubers of ferns can be eaten as fresh greens, cooked as vegetables or processed for starch, and there are about 52 species of edible ferns traditionally consumed as food throughout the country at present [3,4]. In recent years, more and more research has reported the food uses of pteridophytes in different parts of the world.

Ferns are rich in flavonoids [5,6], which applied in the prevention and managing of various diseases such as cancers [7], diabetes [3,8], HIV and so on [9]. *W. japonica* is an important edible and medicinal fern in China, South Korea, Japan and other countries. In a previous report, we found that total flavonoid contents (TFC) of *W. japonica* from different habitats were obviously different. However, so far, the cause of the obvious differences remains unclear.

Climate factors were an important and common factor that affects secondary metabolism, especially flavonoids [10,11,12,13]. Therefore, it is critical to adopt effective analytical methods to understand the response of flavonoid and antioxidant activity of *W. japonica* to climatic factors for species to accomplish their full potential and guide cultivation. 

The Artificial Neural Networks method (ANN) is one of the most widely methods used for studying the response of hydrological factors to climate, multi-objective optimization of material selection for sustainable products, evaluation of an artificial neural network retention index model for chemical structure identification in nontargeted metabolomics and so on [14,15,16]. In recent years, ANN modeling methods have grown in popularity due to their excellent capability to learn and model complex non-linear relationships, widely used in ecology and agronomy. Herein, GAs (genetic algorithms) are one of the most efficient global search methods. A BP network is one of the artificial neural network algorithms, a multi-layer feed forward training method based on error back propagation. 

In this paper, flavonoid and antioxidant activity of *W. japonica* from different habitats were determined with HPLC-ESI-TOF-MS, ANN was used to evaluate the effects of climate factors on flavonoids and antioxidant activities, principal component analysis (PCA) and hierarchical cluster analysis (HCA) were performed, as well as Pearson correlation coefficients were used for verifying the evaluation.

## 2. Results and Discussion

### 2.1. TFC and Antioxidant Activities of W. japonica from Nine Main Production Areas 

As edible and medicinal ferns, TFCs of *W. japonica* have been reported several times, which shows *W. japonica* is rich in flavonoids [5,17,18,19]. In this paper, TFCs of *W. japonica* from nine main producing areas were different, but all were more than 114.55 mg/g in Figure 1. TFC of *W. japonica* from Wuyi Mountain (Jiangxi Province) was the highest (345.4 mg/g), and that of *W. japonica* from Tianmu Mountain was the lowest (114.5 mg/g). 

Antioxidant activities of *W. japonica* from different districts were obviously different. IC50 (including scavenging activities of DPPH·, ABTS·, O_2_^−^·) and reducing force on Fe^3+^ of *W. japonica* from different districts were showed in Figure 2. *W. japonica* from Wuyi Mountain (Jiangxi Province) showed the strongest scavenging to DPPH radicals and reducing force on Fe^3+^, but weaker on eliminating ABTS radicals and superoxide anion radicals. However, *W. japonica* from Fengyang Mountain showed the strongest radical scavenging activity to ABTS radicals and superoxide anion radicals.

TFC and antioxidant activity of plants are two important evaluation indices to selection of plant for healthcare company. Numerous physiological processes and some environmental factors occasionally produce oxygen centered free radicals, which may lead to diseases including cancer, atherosclerosis, cirrhosis and so on [20]. Therefore, the search for antioxidants in industrial plants is a very active field of research [21]. The results in this paper confirmed that *W. japonica* with strong antioxidant activity has the potential to be applied to healthcare companies and deserved further exploiting.

### 2.2. Flavonoids of W. japonica from Different Districts 

By means of HPLC-ESI-TOF-MS, accurate mass, mass match and isotope pattern match contributed to the identification, and eight flavonoids were tentatively identified on the basis of the characteristic fragments present in the spectra and reference spectra available in Figure 3. Rt (retention time), UV_λmax_, the molecular ions and so on are listed in Table 1. Tentatively identified flavonoids contained five flavonols (isotrifolin, rutin, myricetin deoxyhexoside, quercetin-3-rutinoside, quercitrin), two flavones (luteolin 6-C-glucoside and luteolin-4′-*O*-(6″-trans-caffeoyl)-β-d-glucopyranoside) and one isoflavone (genestein G 2).

With UV_λmax_ at 283 nm, molecular ions at *m*/*z* [M-H]^−^463.0882, [M+H]^+^465.1026 and [2M-H]^−^927.1837, C_21_H_20_O_12_ was automatically matched to by the software, and peak 6 was tentatively identified as isotrifolin [22]. Peak 7 with UV_λmax_ at 265, 285 and 350 nm, and molecular ions at *m*/*z* [M-H]^−^609.1461, [M+H]^+^611.1621. C_27_H_30_O_16_ was automatically matched to and tentatively identified as Rutin [23]. With the negative ESI-MS spectrum, peak 9 was automatically matched to a molecular ion at *m*/*z* [M-H]^−^463.0882, the characteristic absorption was at 270, 280 and 355 nm. So peak 9 was tentatively identified as myricetin deoxyhexoside [24]. Peak 10 had UV_λmax_ at 270, 285 and 335 nm, *m*/*z* [M-H]^−^609.1461, was tentatively determined as quercetin-3-rutinoside [25]. Peak 11 had UV_λmax_ at 270, 280 and 340 nm, which was similar to luteolin 6-C-glucoside, molecular ions at *m*/*z* [M-H]^−^447.0933 and automatically matched to C_21_H_20_O_11_ [26]. For peak 12, a molecular ion at *m*/*z* [M-H]^−^447.0933 was observed in negative ESI-MS spectrum, the UV spectrum (245, 265 and 350 nm) which indicated this compound might be flavonol and tentatively identified as quercitrin [27]. The molecular ions of peak 13 were at *m*/*z* [M-H]^−^575.1768 in the negative ESI-MS spectrum, and automatically matched to C_28_H_32_O_13_, with the UV_λmax_ at 235 and 285 nm, it was tentatively identified as genestein G2 [28]. For peak 14, the UV_λmax_ at 235, 280 and 335 nm, and *m*/*z* [M-H]^−^609.0257, so it was tentatively identified as luteolin-4′-*O*-(6″-trans-caffeoyl)-β-d-glucopyranoside [29]. 

The results showed that samples from Longquan were rich in isotrifolin, and samples from Wuyi Mountain (Jiangxi Province) were rich in rutin and myricetin deoxyhexoside. Samples from Tianmu Mountain were rich in quercetin-3-rutinoside, luteolin-4′-*O*-(6″-trans-caffeoyl)-β-d-glucopyranoside and genestein G2, but samples from Wuyuan County were rich in luteolin 6-C-glucoside.

### 2.3. Evaluating for the Effects of Climate Factors on Flavonoids and Antioxidant Activities by the Method of ANN, the Correlation Coefficient Matrix and Hierarchical Cluster Analysis (HCA)

The results of the sensitivity analysis showed that a non-significant difference existed between the climate factors and total flavonoid content, but the effects of climate factors on O_2_^−^ radicals scavenging activity and reducing power of Fe^3+^ of *W. japonica* were more obvious than other antioxidant activities. 

According to the ANN, total flavonoid content was the most sensitive to MIT (Average minimum temperature), followed AP (Average precipitation) and ARH (Average relative humidity) in Figure 4. While O_2_^−^ radicals scavenging activity was the most sensitive to MAT (Maximum temperature) and ARH (Relative humidity), reducing power of Fe^3+^ was the most sensitive to MIT (Average minimum temperature) and AP (Average precipitation), MIT (Average minimum temperature) and AP (Average precipitation) were the main sensitive factors to scavenging activity of DPPH radicals. The key factors affecting ABTS radicals scavenging activity were not obvious, but relatively, low temperature had a slightly larger impact on ABTS radicals scavenging activity in Figure 5. Basically, it showed that low temperature and relative humidity were the key climate factors to influencing total flavonoid content. Decreasing the temperature on a small scale could increase the flavonoid content [30], which explained the above results, while the antioxidant activity was most sensitive to temperature, followed by relative humidity and precipitation.

The Pearson correlation coefficient was used to measure of the strength of the linear relationship between two variables and ranges from −1 to 1 [31]. A linear correlation between the total flavonoid content and antioxidant activity of *W. japonica* and climate factors was listed in Appendix A. The results showed that there was a non-significant difference between the climate factors and total flavonoid content and antioxidant activity of *W. japonica* in Figure 6. However, the influence of climate on the total flavonoids and antioxidant activity did exist. Average daytime precipitation per month and average maximum temperature per month showed a low positive correlation with total flavonoid content, and average sunshine duration per month showed a low negative correlation with total flavonoid content.

Average daytime precipitation per month showed a medium negative correlation with DPPH radical scavenging activity. ABTS radical scavenging activity had a medium positive correlation with maximum temperature and a medium negative correlation with average daytime precipitation. The relationship between O_2_^−^ radical scavenging activity and average relative humidity was a medium negative correlation. In addition, the low positive correlation between average maximum temperature and the reducing force on Fe^3+^ was detected. 

HCA was performed with a heatmap visualizing to illustrate the differences among samples. According to HCA in Figure 7, the total flavonoid content and antioxidant activity were more affected by average relative humidity and sunshine duration than other climate factors. Among the nine main producing areas, the areas with higher average overnight precipitation went along with relatively higher total flavonoids content, stronger scavenging DPPH radical activity and reducing power of Fe^3+^. However, the producing area with the highest average temperature per month had higher scavenging activity of ABTS radicals and O_2_^−^ radicals.

The above results showed that there was a non-significant difference between the climate factors and total flavonoid content, but temperature, precipitation and relative humidity did affect the total flavonoid content of *W. japonica*. Antioxidant activity was confirmed to be relative to precipitation, followed by temperature. Through the statistical validation, the evaluating by the method of ANN for the effects of climate factors on flavonoids and antioxidant activities was accurate and reasonable.

### 2.4. Response of Flavonoid Type of W. japonica to Climatic Factors with ANN, Pearson Correlation Coefficient, PCA and HCA

The results of the sensitivity analysis (ANN) showed that sunshine duration was the main and key influential factors on flavonoid type of *W. japonica* in Figure 8. Besides sunshine duration, temperature was also effective influential factors on quercetin-3-rutinoside, luteolin 6-C-glucoside, quercitrin, genestein G2, luteolin-4′-*O*-(6″-trans-caffeoyl)-β-d-glucopyranoside.

In addition, genestein G2 and luteolin-4′-*O*-(6″-trans-caffeoyl)-β-d-glucopyranoside were also sensitive to average precipitation. Based on this, it was evaluated that flavones and flavonols were most sensitive to sunshine and temperature, but isoflavone was sensitive to sunshine, temperature and precipitation. In addition, the accumulation of rutin, myricetin deoxyhexoside, quercetin-3-rutinoside and quercitrin would be also affected by average maximum temperature.

The Pearson correlation coefficients were listed in Appendix A. Quercetin-3-rutinoside showed high negative correlation with average minimum temperature per month, and medium negative correlation with average precipitation per month. Luteolin-4′-*O*-(6″-trans-caffeoyl)-β-d-glucopyranoside was moderately negatively correlated with average precipitation. Myricetin deoxyhexoside and quercitrin were moderately negatively correlated with average sunshine duration. Isotrifolin was moderately positively correlated with average minimum temperature and average precipitation, but moderately negatively correlated with average sunshine duration. Rutin and luteolin 6-C-glucosidewere both moderately negatively correlated with average minimum temperature. Genestein G2 was positively correlated with average sunshine duration and relative humidity. In a word, the content of isoflavone was positively correlated with average sunshine duration and relative humidity. The content of flavonol was related to sunshine duration, overnight precipitation and low temperature.

Principal component analysis (PCA) was a common mathematical procedure, which was inclined to reduce high-dimensional data by expressing them with a set of linearly uncorrelated orthogonal basis. According to PCA score plots in Figure 9, it was found that PC1 mainly reflects the information of average precipitation per month, average minimum temperature per month and overnight precipitation, and PC2 mainly reflects the information of daytime precipitation per month. The score of isotrifolin and quercetin-3-rutinoside (flavonol) on PC1 was higher than other compounds. Therefore, it could be speculated that the content of isotrifolin and quercetin-3-rutinoside (flavonol) were mainly influenced by overnight precipitation and minimum temperature, especially for the content of isotrifolin.

Conversely, the score of rutin, myricetin deoxyhexoside, quercitrin, luteolin-4′-*O*-(6″-trans-caffeoyl)-β-d-glucopyranoside on PC2 was higher than other compounds. It followed that the content of rutin, myricetin deoxyhexoside, quercitrin, luteolin-4′-*O*-(6″-trans-caffeoyl)-β-d-glucopyranoside was mainly affected by daytime precipitation, especially for the content of myricetin deoxyhexoside.

The dendrogram and heatmap of hierarchical cluster analysis clearly and visually show the influence of climate factors on different flavonoid type in Figure 10. The positive influence of overnight precipitation and minimum temperature on the content of isotrifolin was obvious, but excessive sunshine duration might inhibit the producing of isotrifolin. The content of rutin was related to sunshine duration. Main factors influencing the content of myricetin deoxyhexoside were overnight precipitation, average relative humidity and sunshine duration. Average relative humidity was the key climate factor of genestein G2 (isoflavone). The contents of quercetin-3-rutinoside and luteolin 6-C-glucoside might be reduced by low temperature and overnight precipitation. The effect of maximum temperature on quercitrin was significant, but excessive sunshine could inhibit the producing of quercitrin. luteolin-4′-*O*-(6″-trans-caffeoyl)-β-d-glucopyranoside mainly be positive impacted by sunshine duration and relative humidity, but negative impacted by overnight precipitation.

The above results showed that flavonol and isoflavone in the tentatively identified components, except for quercetin-3-rutinoside, were mainly affected by sunshine duration, and flavones were probably more related to temperature. Through the statistical validation, the evaluating by the method of ANN for the effects of climate factors on flavonoids type was basically consistent with our results except for quercetin-3-rutinoside. 

Secondary metabolites were considered to be related to abiotic factors [32,33]. Sunshine had certain impact on plant morphological, plant development and metabolites [11]. Combined with the above analysis results, it was found that sunshine duration did have a great influence on flavonols and isoflavone. The result further confirmed the effect of sunshine on plant metabolism. 

It was reported that temperature was an important factor affecting flavonol accumulation [34], low temperature would promote the accumulation of flavonols [35], and during 30–40 °C, flavonol synthesis was inhibited [36,37]. In this result, the accumulation of rutin, myricetin deoxyhexoside, quercetin-3-rutinoside and quercitrin (flavonols) were relative to average maximum temperature per month. However, low temperature could not promote the accumulation of all flavonol compounds. Only the accumulation of isotrifolin was found to be significantly relative to low temperature in this paper.

A little information was reported on the effects of relative humidity on the contents of flavonol in fruits and berries. Yang et al. [38] reported that flavonol glycosides of red currants were little relative to humidity variables. In this paper, relative humidity was negative correlation to O_2_^−^ radical scavenging activity.

At present, no significant correlation was observed between the precipitation and the contents of each secondary metabolite. Kalinova et al. [39] reported that no correlations between epicatechin (flavanol) and precipitation were statistically significant. Ribeiro et al. [40] indicated that precipitation could reflect a stronger in vitro antioxidant activity of Secondatia floribunda. Dong et al. [41] found that there seems to be no direct relationship between the flavonoid content of *Eucommia ulmoides* and the precipitation, which was consistent with our result. In this paper, it was elucidated that the precipitation was no significantly associated with flavonoids, but significantly affected antioxidant activity. Through building ANN, the effects of climate factors on flavonoids and antioxidant activities were evaluated, and the results have been verified to be effective.

## 3. Materials and Methods

### 3.1. Plant Materials

*W. japonica* as a represent species in Blechnaceae, distributed in the south of the Yangtze River in China. The height of *W. Japonica* is 0.8–1.2 m. The rhizome is stout, recumbent and dark. The sporangia group is linear, on both sides of the main vein narrow long mesh in Figure 11. 

Nine main production areas of *W. japonica* were showed in Figure 12. Meteorological data (August in 2015) were derived from National Meteorological Information Center, China Meteorological Administration. Temperature, precipitation, sunshine duration and relative humidity were collected from nine climate stations (Shangrao, Hongjia, Yushan, Yuhuan, Qimen, Lishui, Liuhe, Jianyang, Jianyang and Wuyi Mountain) Appendix A. About 5–10 samples were randomly selected from each sample site. Marked specimens (SNU-2015-8) were stored in College of Life science, Shanghai Normal University.

### 3.2. Chemicals

Rutin (purity > 99.0%), DPPH (2, 2-diphenyl-1-picrylhydrazyl), ABTS (2, 21-azinobis-(3-ethylbenzothiazoline-6-sulfonic acid), NBT (Nitrotetrazolium blue chloride), PMS (Phenazine methosulfate), NADH (Nicotinamide adenine dinucleotide) and TPTZ (2,4,6-tri-2-pyridyl-s-triazine), were purchased from Sigma Co. (Shanghai, China).

### 3.3. Preparation of Plant Extracts

The whole plants of *W. japonica* were separately placed in the shade for 72 h, and then dried in an oven for 48 h (75 °C) and crushed. After filtering through a 40-mesh screen, collected respectively. One gram powder of the sample was respectively extracted twice with 60% ethanol, and the constant volume was 50 mL. The extract was stored at 4 °C respectively. 

### 3.4. Determination of TFCs

The methods for determination of flavonoids was similar to previous reports [42]. The concentration of rutin standard solution was 20 µg/mL. The calibration curves were y = 10.655x − 0.0076 and R^2^ = 0.99869. Absorbance was read on a TU-1810 UV-spectrophotometer (Beijing, China) at 510 nm.

Step 1: With the method of modified colorimetric assay, rutin was taken as the standard. By the way of drawing standard curve, 5 mL different concentration rutin (0.02 mg/mL, 0.04 mg/mL, 0.06 mg/mL, 0.08 mg/mL, 0.1 mg/mL) was added 0.3 mL 5% NaNO_2_ (6 min), 0.3 mL 5% Al(NO_3_)_3_ (6 min), 4.4 mL 4% NaOH (12 min) in turn, and then optical density (OD) at 510 nm were recorded respectively. According to the rutin standard curve, the linear equation (y = Bx + A) of one variable is synthesized. 

Step 2: 2 mL extract from different collecting sites was also added NaNO_2_, Al(NO_3_)_3_ and NaOH in turn, the method was same with the step 1. The formula used was as follows:TFC (%) = [(OD_1_ + OD_2_ + OD_3_)]/3 − A]/B × 10/2 × volume/1000 × 100%(1)

### 3.5. Antioxidant Activity

#### 3.5.1. DPPH· Scavenging Assay

Briefly, 1 mL 0.1 mM DPPH and 1 mL different concentrations of 60% ethanol extract of *W. japonica* were mixed and then incubated for 30 min in the dark [43]. The absorbance was recorded at 517 nm. Methanol as the control was substituted for sample. The scavenging percentage (%) was calculated by the following equation (All the determinations were performed in triplicate and found to be reproducible within the experimental error):DPPH scavenging percentage (%) = (1 − A_sample517_/A_control517_) × 100(2)

#### 3.5.2. ABTS· Scavenging Assay

The mixture (7 mM ABTS and 2.45 mM potassium persulfate) was in the dark for 12 h before use. When the absorbance at 734 nm reached to 0.7 ± 0.01, different concentration extracts were mixed with ABTS solutions for 6 min [43]. The absorbance at 734 nm was recorded, and 60% methanol was taken as the control. The scavenging activity (%) was calculated by the following equation (all the determinations were performed in triplicates and found to be reproducible within the experimental error):ABTS scavenging activity (%) = (1 − A_sample734_/A_control734_) × 100(3)

#### 3.5.3. Superoxide Anion (O_2_^−^) Scavenging Assay

One mL of NBT (150 µM), 1.0 mL of NADH (468 µM) dissolved in sodium phosphate buffer (100 mM, pH = 7.4) and 1.0 mL of PMS (60 µM) dissolved in purified water were added with 1.0 mL different concentration extracts, respectively [42]. After incubating at 25 °C for 5 min, the absorbance at 560 nm was recorded. The scavenging activity (%) was calculated by the following equation (all the determinations were performed in triplicates and found to be reproducible within the experimental error):Superoxide anion scavenging activity (%) = (1 − A_sample560_/A_control560_) × 100(4)

#### 3.5.4. Reducing Force on Fe^3+^ Assay

One mL extract of different concentrations dissolved in 2.5 mL phosphate buffer (pH = 6.0) mixed with 2.5 mL of potassium ferricyanide (1%, *w*/*v*) at 50 °C. The reaction was terminated with 10% TCA after 20 min. The mixture centrifuged at 3000 rpm for 10 min were divided into two parts, one part (part A) of which were added 2.5 mL of distilled water and 0.5 mL of 0.1% ferric chloride in turn, and the other part (part B) mixed with 3 mL of distilled water. The absorbance was read at 700 nm, respectively. Part B was taken as blank [42]. (All the determinations were performed in triplicates and found to be reproducible within the experimental error).

### 3.6. HPLC-ESI-TOF-MS Analysis

The conditions of chromatographic gradient separation and the gradient elute settings of mobile phase were as same as previous paper [43].

### 3.7. Model Construction

#### 3.7.1. Sample Organization

Measured parameters/sample variables include: average temperature per month, maximum temperature per month, minimum temperature per month, average precipitation per month, daytime precipitation per month, overnight precipitation per month, sunshine duration per month, average relative humidity per month. The effect of each observed parameter on the content of compound 1–8 was evaluated. Subsequently, samples include environmental factors were organized on the basis of the GAs. Meanwhile, the key factors affecting the content of compound 1–8 were identified.

#### 3.7.2. The Selection of Parameter

On the basis of an improved BP algorithm, network training combined the optimized initial weight of the GAs with the limited supervised adjustment of learning rates [44]. In order to improve the BP net learning algorithm avoiding local defects, BP net was optimized by the genetic algorithm. Firstly, optimizing the weights and threshold of BP net, and then assigned to optimized BP net. This method can effectively avoid the original local defects of BP net, meanwhile with the characteristics of fast convergence and high accuracy. The hidden layer node of the BP net is determined as 30–20 under multiple trials. An S-type logsig function was used for transferring the net, the topology structure of which was 8-50-30-1. All the input of the samples was between 0.10 and 0.90.

All the assays were repeated three times, and the average output error was less than 0.02. The error estimation formula to be:ε < (Δ)^2^ × N (where N is sample size)(5)
where N is sample size.

#### 3.7.3. Sensitivity Analysis

Seven environmental factors influence on nine flavonoids content were analyzed and discussed with GAS.

### 3.8. Statistical Analysis

After normalization of the data, PCA and HCA as well as the Pearson correlation coefficients were used for statistical analysis. The mean of three independent experiments was recorded, and the main statistical software was R-project (Auckland, New Zealand), Origin 7.5 (Northampton, MA, USA) and matlab 7.0 (Natick, MA, USA).

## 4. Conclusions

In current research, the differences of total flavonoid contents and antioxidant activities of *W. japonica* from different regions were significant. *W. japonica* from Wuyi Mountain (Jiangxi Province) showed the highest total flavonoid contents (345.4 mg/g) and the strongest scavenging to DPPH radicals and reducing force on Fe^3+^. However, *W. japonica* from Fengyang Mountain showed the strongest radical scavenging activity to ABTS radicals and superoxide anion radicals. In addition, five flavonols, two flavones and one isoflavone were tentatively identified in *W. japonica* with HPLC-ESI-TOF-MS. Further results showed that the response of flavonoids and antioxidant activity to climatic factors was complex, combined ANN and statistical analysis, but it was certain that sunshine duration was the key factor on flavonol and isoflavone, and temperature was the key factor on flavones. All of the above results illustrate that the effects of climate factors on flavonoids and antioxidants was significant, which would provide an important basis for further exploring the mechanism of climate affecting secondary metabolites.

## Figures and Tables

**Figure 1 molecules-28-01985-f001:**
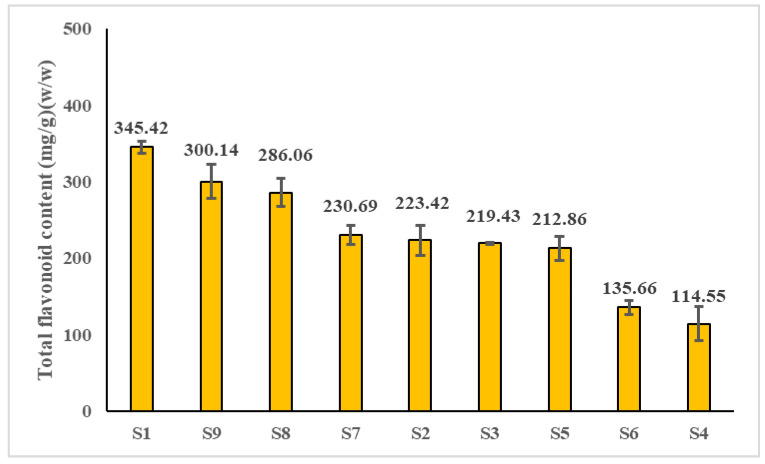
TFCs of *Woodwardia japonica* from different districts (mg/g) (*w*/*w*).

**Figure 2 molecules-28-01985-f002:**
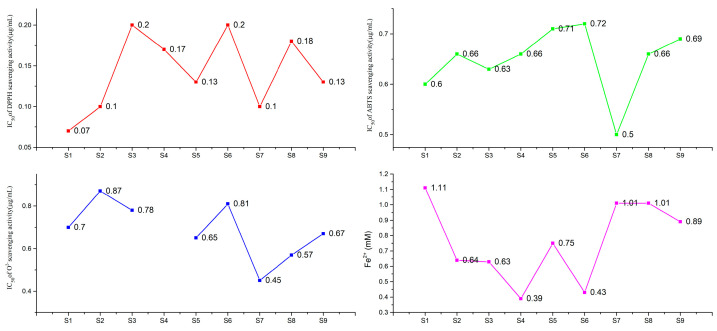
IC_50_ (including scavenging activities of DPPH, ABTS, O_2_^−^) and reducing force on Fe^3+^ of *W. japonica*.

**Figure 3 molecules-28-01985-f003:**
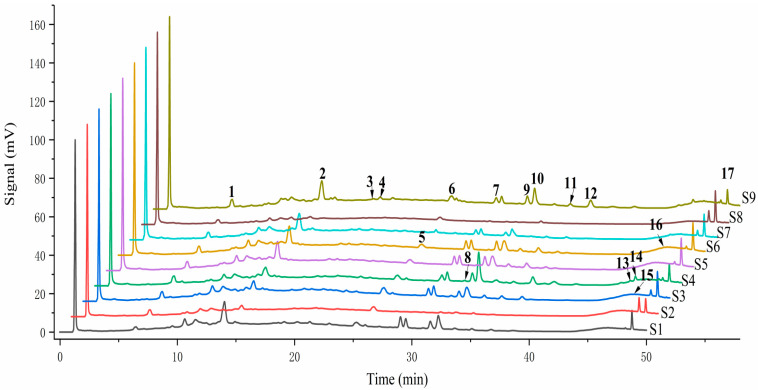
Flavonoids of *W. japonica* from different habitats (Sample sites shown in Appendix A).

**Figure 4 molecules-28-01985-f004:**
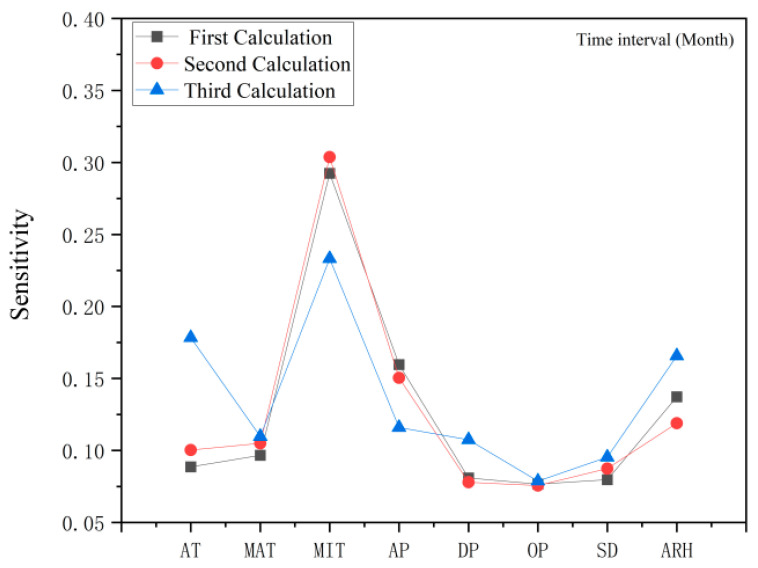
Response of total flavonoid content to climate factors based on the Artificial Neural Network Model. AT: Average temperature per month; MAT: Maximum temperature per month; MIT: minimum temperature per month; AP: Average precipitation per month; DP: Daytime precipitation per month; OP: Overnight precipitation per month; SD: Sunshine duration per month; ARH: Average relative humidity per month.

**Figure 5 molecules-28-01985-f005:**
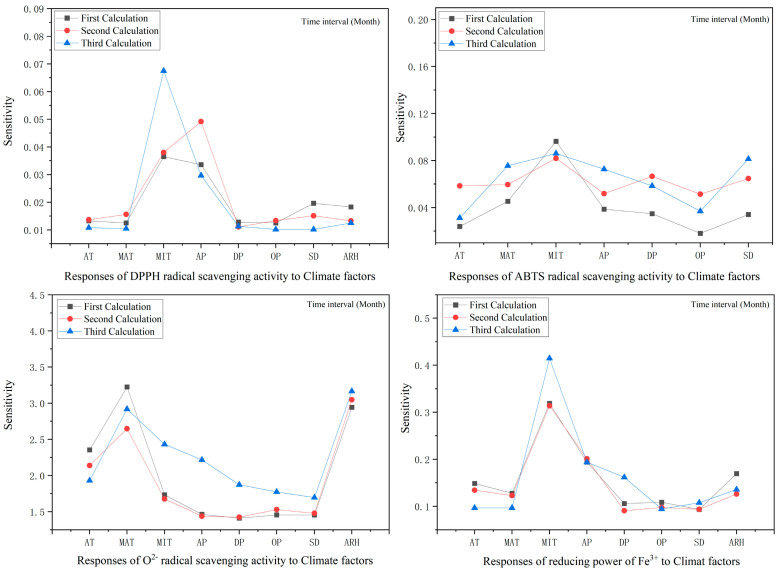
Response of antioxidant activity to climate factors.

**Figure 6 molecules-28-01985-f006:**
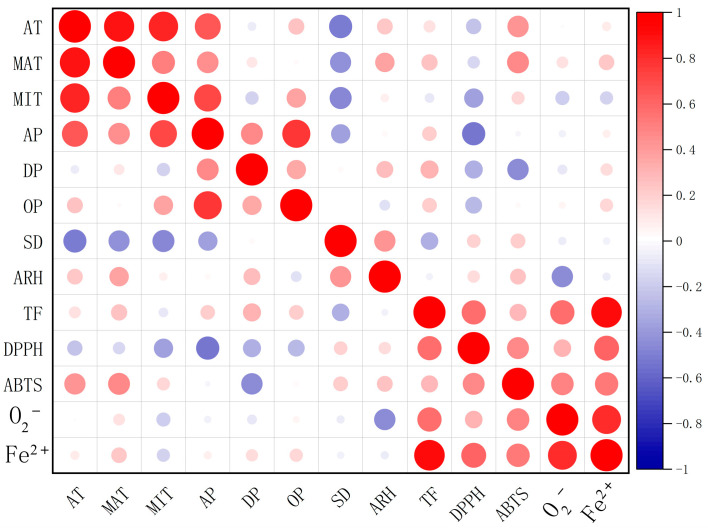
The correlation coefficient matrix between TFC, antioxidant activity of *W. japonica* and climate factors.

**Figure 7 molecules-28-01985-f007:**
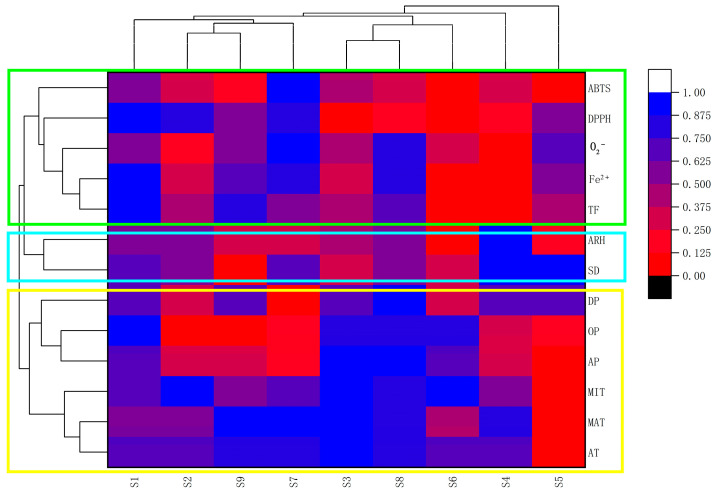
Dendrogram and heatmap of hierarchical cluster analysis of climate factors, TFC and antioxidant activity among nine main producing areas of *W. japonica.* Concentrations are illustrated by a color gradient from blue (high) to red (low).

**Figure 8 molecules-28-01985-f008:**
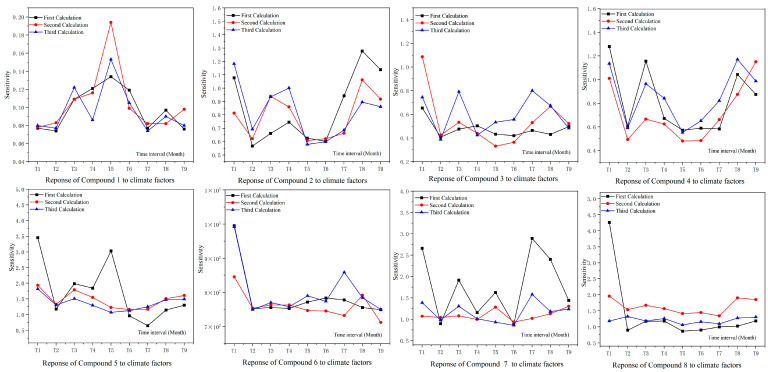
Compound 1–8: isotrifolin, rutin, myricetin deoxyhexoside, quercetin-3-rutinoside, luteolin 6-C-glucoside, quercitrin, genestein G2, luteolin-4′-*O*-(6″-trans-caffeoyl)-β-d-glucopyranoside.

**Figure 9 molecules-28-01985-f009:**
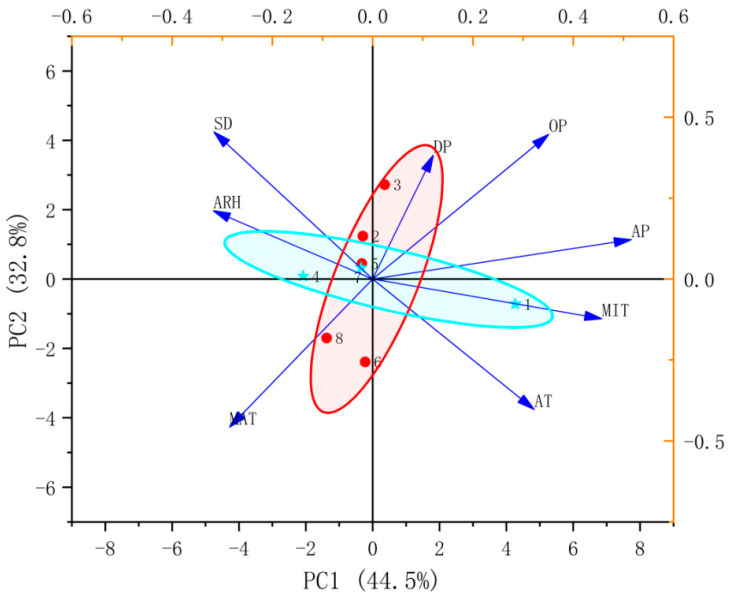
PCA score plots using flavonoid types with different climate factors (*n* = 3). 1–8: isotrifolin, rutin, myricetin deoxyhexoside, quercetin-3-rutinoside, luteolin 6-C-glucoside, quercitrin, genestein G2, luteolin-4′-*O*-(6″-trans-caffeoyl)-β-d-glucopyranoside. The blue arrows refer to different climate factors.

**Figure 10 molecules-28-01985-f010:**
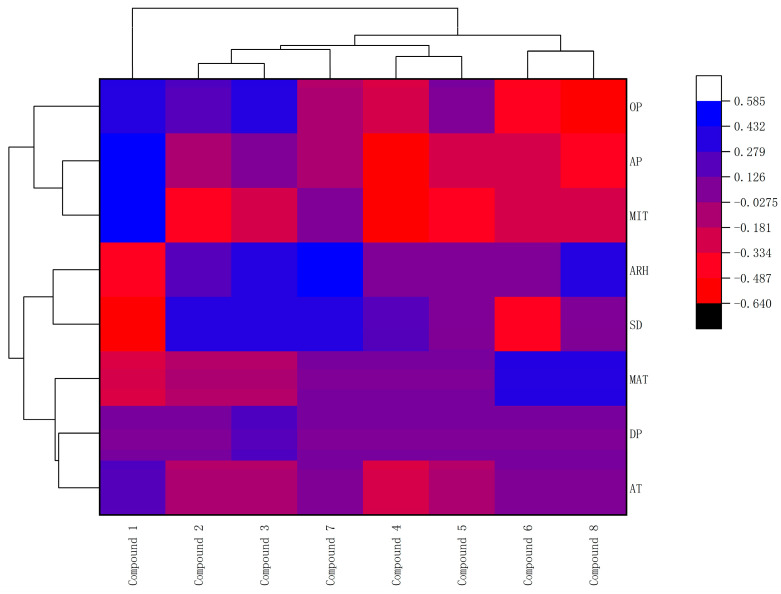
Dendrogram and heatmap of hierarchical cluster analysis of climate factors among eight flavonoids of *W. japonica.* Concentrations are illustrated by a color gradient from blue (high) to red (low). Compound 1–8: isotrifolin, rutin, myricetin deoxyhexoside, quercetin-3-rutinoside, luteolin 6-C-glucoside, quercitrin, genestein G2, luteolin-4′-*O*-(6″-trans-caffeoyl)-β-d-glucopyranoside.

**Figure 11 molecules-28-01985-f011:**
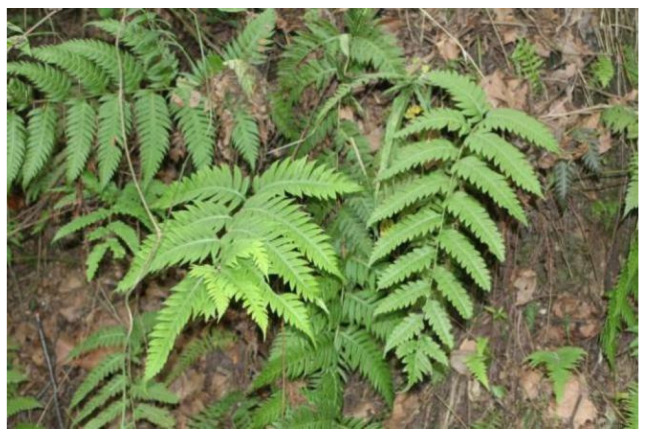
Morphological characteristics of *W. japonica*.

**Figure 12 molecules-28-01985-f012:**
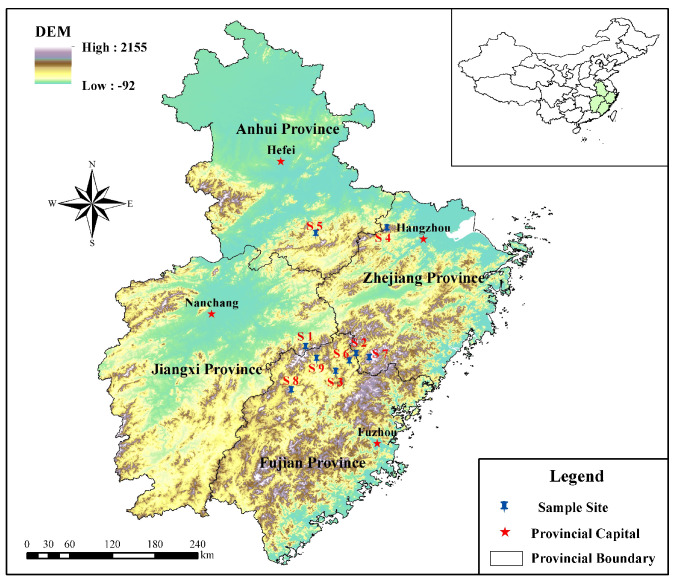
The distribution of nine main producing areas of *W. japonica*.

**Table 1 molecules-28-01985-t001:** Qualitative Analysis of Compounds of *Woodwardia japonica* with HPLC-ESI-TOF-MS.

No.	RT(Min)	Compound Formula	Molar Mass	Experimental *m*/*z*	Error		UV (nm) λmax	Identification
					(ppm)	(mDa)		
1	6.74	C_7_H_6_O_4_	154.0269	[M-H]^−^153.0193, [M+H]^+^155.0394	−1.59	−0.24	280	Protocatechuate
2	14.76	C_16_H_18_O_9_	354.0954	[M-H]^−^353.0877, [2M-H]^−^707.1832 [M+H]^+^355.1027	−0.79	−0.28	285,325	Chlorogenic acid
3	19.45	C_32_H_36_O_16_	676.1999	[M-H]^−^675.1931, [M-C_7_H_12_O_2_]^−^547.1466 [M-C_7_H_12_O_2_-C_10_H_10_O_5_]^−^338.0996	0.69	0.47	283,313	Unknown
4	21.50	C_16_H_16_O_8_	336.0845	[M-H]^−^335.0772	0.17	0.06	230,280	Unknown
5	25.04	C_18_H_26_O_8_	370.1626	[M-H]^−^369.1555	0.5	0.19	230,285	Unknown
6	25.87	C_21_H_20_O_12_	464.0954	[M-H]^−^463.0882, [M+H]^+^465.1026 [2M-H]^−^927.1837	−0.22	−0.1	283	Isotrifolin (flavonol)
7	29.58	C_27_H_30_O_16_	610.1527	[M-H]^−^609.1461, [M+H]^+^611.1621	1.16	0.71	265,285,350	Rutin (flavonol)
8	31.17	C_33_H_48_O_16_	700.2945	[M-H]^−^699.287	−0.42	−0.29	230,280	Unknown
9	31.7	C_21_H_20_O_12_	464.0954	[M-H]^−^463.0882, [2M-H]^−^827.1829 [M+H]^+^465.1041	0.08	0.04	270,280,355	Myricetin deoxyhexoside (flavonol)
10	32.35	C_27_H_30_O_16_	610.1533	[M-H]^−^609.1461, [M+H]^+^611.1613	0.15	0.09	270,285,335	Quercetin-3-rutinoside (flavonol)
11	35.28	C_21_H_20_O_11_	448.1011	[M-H]^−^447.0933, [M+H]^+^449.1095 [2M-H]^−^895.1918	−1.26	−0.57	270,280,340	Luteolin 6-C-glucoside (flavone)
12	37.44	C_21_H_20_O_11_	448.101	[M-H]^−^447.0933, [2M-H]^−^895.1927 [M+H]^+^449.1094	−1.02	−0.46	240,265,350	Quercitrin (flavonol)
13	45.69	C_28_H_32_O_13_	576.1841	[M-H]^−^575.177, [M+H]^+^577.1912	0.38	0.22	235,285	Genestein G 2 (isoflavone)
14	46.16	C_30_H_26_O_14_	610.1326	[M-H]^−^609.125, [M+H]^+^611.1388	−0.57	−0.35	235,280,335	Luteolin-4′-*O*-(6″-trans-caffeoyl)-β-d-glucopyranoside (flavone)
15	46.59	C_20_H_36_O_12_	468.2209	[M-H]^−^467.2134	−0.58	−0.27	235,280	Unknown
16	47.84	C_22_H_38_O_12_	494.2372	[M-H]^−^493.2291	−1.8	−0.89	238,280	Rhodioloside B
17	49.04	C_28_H_46_O_8_	510.3189	[M-H]^−^509.312	0.71	0.36	245,280	Pladienolides F/Pladienolides G

## Data Availability

Data are available from the corresponding author on requested.

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
