# Peer review of "Elucidating Flavonoid and Antioxidant Activity in Edible and Medicinal Herbs Woodwardia japonica (L.f.) Sm. Based on HPLC-ESI-TOF-MS and Artificial Neural Network Model: Response to Climatic Factors"

_molecules, 2023, doi:10.3390/molecules28041985_

Round 1

Reviewer 1 Report (Previous Reviewer 3)

The article has been corrected but still contains serious flaws.

abstract: TFC should be explained, the abstract should contain information about the methods used to measure flavonoid and antioxidant activity, 1-2 sentences of conclusions should be added in the abstract

lines 38–40 are the results rather than the introduction

lines 59, 139, 205 – the abbreviation ANN has already been introduced before - no need to explain

line 64 – TFC should be explained

lines 74, 141, 177, 188, 287, 353 – superoxide anion radical records are still not correct

line 296 – should be ANN instead of artificial neural networks

line 342 – add information about the absorbance reader used (model, manufacturer, country)

line 397 – complete information about the statistical software used (version, manufacturer, country)

conclusions should be supplemented with information on the practical application of the obtained results

Author Response

Reviewer 2 Report (New Reviewer)

Line 38.   In this paper...345 mg... This result of the current work should not be placed in introduction.

Line 45. Rewrite sentence

Figure 2. To improve resolution of the figure

The quantification of flavonoids should be clarified in materials and methods, standards used to quantify, calibration curves, LOD, LOQ of the method.

Table 1. Check compound 11 identification. Do you mean luteoline glycoside? Because the mass and formula do not match with luteolin.

Line 107. Isomer 2? There is no isomer 1.

Line 110. Flavone? Correct to flavonol.

 -Quantities of identified flavonoids should be included in a table in the manuscript.

Do the authers have data about altitudes of different locations? Would that affect the results?

Round 2

Reviewer 1 Report (Previous Reviewer 3)

 line 22 – should be „The results showed 1) total flavonoid content (TFC)” instead of „The results showed 1) TFC (total 22 flavonoid content)”

The last sentence of the abstract should be a summary of the results. Currently, these conclusions do not match the results obtained.

line 81 – superoxide anion radical records are still not correct

Author Response

This manuscript is a resubmission of an earlier submission. The following is a list of the peer review reports and author responses from that submission.

Round 1

Reviewer 1 Report

The article shows a characterization of the flavonoid composition of the fern Woodwardia japonica in terms of its major components and its antioxidant activity. In addition, an attempt has been made to investigate which environmental variable explains a higher concentration of these compounds by means of a global sensitivity analysis. 

The document needs to be thoroughly revised since both the language used, and the information shown in some parts need to be improved.

One of my main concerns is that the study should have been carried out over a sufficiently long period of time to draw well-founded conclusions (i.e., a kind of a longitudinal study). The composition of the biological material analysed will show a significant variability depending on environmental stressors. Hence before reaching conclusions the results need to be confirmed with samples taken at other times to average the impact of uncontrolled variables. This would provide greater strength to the conclusions obtained. In other words, can the results be extrapolated to other moments in the year? Or even to other years? Without this discussion the conclusion regarding which geographic zone is the best for collecting the studied specie is not clearly supported.

There are expressions that appear throughout the manuscript that seem to indicate that the paper has not been carefully prepared. For example, in the introduction, the expressions rich in high nutritional content (line 31) and metabolites of health care (line 32) are used. The meaning of these expressions is not clear (they are only a few examples) and they make it difficult to understand the document.

 In addition, the headings of Tables 1 and 3 are the same and refer to aspects that have not been addressed in the document.

 On the other hand, ecological data are shown in Table 3, however, there is no indication of how these data were obtained or even the source of the data. In the case of relative humidity, there is no indication of what this value represents (i.e. mean).  

Section 3.4 describes the methodology used to determine the total flavonoid content. Can the authors explain what the values of OD1, OD2 and O3 refer to? This is not clear from the paper.

Reviewer 2 Report

The study involved effects of different habitats on flavonoid and antioxidant activity of edible and folk medicinal fern Woodwardia japonica. Antioxidant activities were evaluated using different approaches, while flavonoids identification was tentative using HPLC-ESI-TOF-MS. Although the aspects of this work are of reader’s scientific interest, generally it requires more scientific elaboration. In addition, this manuscript requires proofreading by the English native speaker.

Line 37. Reference 8 and 9 are not good for supporting statement of 16 isolated compounds, specially ref.  9 which is only Congress Abstract. Please give more info of W. japonica compound identification. In addition, please check and correct the name of the compounds in all the text (e.g. luteolin-4'-O-(6"-trans-caffeoyl)-β-D-glucopyranoside.

Line 48. and in 2.4. Please elaborate GSA background used in this study (some previous example approach would be sufficient).

Tables 1, 2 and 3 titles is out of scope of the manuscript.

2.2. Why did the authors use DPPH and ABTS assays – as they both use the same approach ie. to check radical scavenging ability?

Line 84 Rutin as tentative? The authors stated that they had standard (see 3.2)

2.4. Qualitative informations of flavonoids are provided, but not quantitative. So, statement that W. japonica is rich in flavonoids (here as well as in the conclusion) should be more elaborated.

3.1. and Table 3. In addition, please provide the informations when the  samples were collected, and how are the information (such as AMT, SD; EAT, AAR, RH) obtained.

3.6. Please report the info, as the ref. 41 is not by the listed authors.

Reviewer 3 Report

Article entitled “Effects of different habitats on flavonoid and antioxidant activity of edible and folk medicinal fern Woodwardia japonica” presented flavonoids and antioxidant properties of W. japonica from nine regions.

I advise the authors to find a native English speaker to proofread the manuscript. The article contains a lot of typos, as well as unfinished sentence (e.g. line 32).

Line 49 – BP should be explained

Material and methods ‒ When were samples of W. japonica collected from different areas? The timing of collection may affect the results.

Line 182 – incorrect chemical record of superoxide anion

Material and methods – incorrect record of pH

Authors use too many abbreviations without explanation, eg NBT or PMS

More detailed description of the statistical methods used should be added

Results of total flavonoid contents or antioxidant activities should be supplemented with a statistical analysis

Section 2.1 requires discussing the results with the literature

Authors in the materials and methods mention that they used the FRAP method, but there are no results.

The article requires an extensive discussion. The description of the use of the W. japonica, e.g. in medicine, can be extended.

To sum up, the article requires a thorough revision.